# Quantification of Mitochondrial Oxidative Phosphorylation in Metabolic Disease: Application to Type 2 Diabetes

**DOI:** 10.3390/ijms20215271

**Published:** 2019-10-24

**Authors:** Matthew T. Lewis, Jonathan D. Kasper, Jason N. Bazil, Jefferson C. Frisbee, Robert W. Wiseman

**Affiliations:** 1Department of Physiology, Michigan State University, East Lansing, MI 48824, USA; lewism29@msu.edu (M.T.L.); kasper249@gmail.com (J.D.K.); jnbazil@msu.edu (J.N.B.); 2Present address: Molecular Physiology Institute, Duke University, Durham, NC 27701, USA; 3Department of Medical Biophysics, University of Western Ontario, London, ON N6A 3K7, Canada; jfrisbee@uwo.ca; 4Department of Radiology, Michigan State University, East Lansing, MI 48824, USA

**Keywords:** phosphate potential, ATP free energy, oxygen delivery, exercise, muscle performance, aerobic capacity, metabolic disease, insulin resistance

## Abstract

Type 2 diabetes (T2D) is a growing health concern with nearly 400 million affected worldwide as of 2014. T2D presents with hyperglycemia and insulin resistance resulting in increased risk for blindness, renal failure, nerve damage, and premature death. Skeletal muscle is a major site for insulin resistance and is responsible for up to 80% of glucose uptake during euglycemic hyperglycemic clamps. Glucose uptake in skeletal muscle is driven by mitochondrial oxidative phosphorylation and for this reason mitochondrial dysfunction has been implicated in T2D. In this review we integrate mitochondrial function with physiologic function to present a broader understanding of mitochondrial functional status in T2D utilizing studies from both human and rodent models. Quantification of mitochondrial function is explained both in vitro and in vivo highlighting the use of proper controls and the complications imposed by obesity and sedentary lifestyle. This review suggests that skeletal muscle mitochondria are not necessarily dysfunctional but limited oxygen supply to working muscle creates this misperception. Finally, we propose changes in experimental design to address this question unequivocally. If mitochondrial function is not impaired it suggests that therapeutic interventions and drug development must move away from the organelle and toward the cardiovascular system.

## 1. Introduction

### Definitions

A brief list of specific definitions is provided for clarity in conceptual aspects of this document. They are to be used in place of other meanings that may have several less stringent descriptions within the literature.

Mitochondrial respiratory capacity (f_MITO_)—Functional capacity inherent to the organelle. Equivalent to mitochondrial function.

Mitochondrial function—Mitochondrial function is the ability of the mitochondria to sense and match the ATP demand [1]. Increased function therefore means increased ATP production capacity per organelle and conversely decreased function means decreased ATP production capacity per organelle.

Mitochondrial fractional volume (V_MITO_)—An indicator for the amount of mitochondria within a tissue. V_MITO_ is the amount of volume mitochondria hold relative to the volume of the cell. This definition does not discriminate between hypotheses on mitochondrial networks or individual organelles. This measure is derived from the estimates of mitochondrial protein, respiratory enzyme activity, and other biochemical determinations (see Larsen et al. [2]).

Muscle mitochondrial oxidative phosphorylation capacity (MOP Capacity)—Absolute capacity for muscle to produce ATP aerobically. This is a function of the V_MITO_ and f_MITO_: $$MOP\;capacity=\;V_{MITO}\;\times \;f_{MITO}\;$$
MOP capacity is often what is measured in vivo in whole tissue.

Fasting blood glucose (FBG)—Refers to 12 h fasting conditions. 

Nicotinamide adenine dinucleotide—reduced form, NADH; oxidized form NAD^+^

Flavin adenine dinucleotide—reduced form, FADH_2_; oxidized form FAD

Phosphate potential—Defined by the free energy of ATP hydrolysis (ΔG_ATP_) at 37 ^°^C, pH 7.0. Its magnitude increases further from equilibrium when the $\frac{\left[ADP\right]\left[Pi\right]}{\left[ATP\right]}$ ratio decreases: $$\Delta G_{ATP}=\;\Delta G_{ATP}^{o}+2.58\;\times \ln(\frac{\left[ADP\right]\left[Pi\right]}{\left[ATP\right]})$$

The more negative the ΔG_ATP_ (kj/mol) the greater the phosphate potential and in healthy muscle cells at rest ranges from about −64 to −68 kj/mol at rest and decreases with intense exercise but not below −48 kj/mol where the energy required for calcium pumping by the sarcoplasmic reticulum can no longer be met resulting in fatigue [3].

Redox potential—Determined from the reduction status of the pyridine pool the redox potential describes the energy contributed to the proton motive force from NADH oxidation at 37 ^°^C, pH 7.0: $$\Delta G_{REDOX}=\;\Delta G_{REDOX}^{o}+2.58\;\times \ln(\frac{\left[NAD^{+}\right]}{\left[NADH\right]})$$

The more negative ΔG_REDOX_, the greater the redox potential and this value ranges from roughly −210 to −230 kj/mol when near full oxidation or full reduction respectively. Note that up to three ATP are produced from the oxidation of one NADH molecule when optimally coupled.

Metabolic diseases present with deficient handling of glucose and lipids and developing comorbidities that ultimately lead to premature death [4]. The most common and well-studied example of metabolic disease is type 2 diabetes which is a growing epidemic with 30.3 million patients in the United States in 2015 [5] and is expected to exceed 54 million by 2030 [6]. Type 2 diabetes is characterized by increased blood glucose, resistance to insulin action, and beta cell failure [7]. This ultimately results in reduced net glucose disposal and increased risk for cardiovascular disease, blindness, renal failure, nerve damage, loss of limb, and other co-morbidities [8,9]. Exercise intolerance, defined as a susceptibility to fatigue or reduced exercise capacity (VO_2Max_), has also been a disputed co-morbidity of this disease [10,11,12,13,14,15,16,17]. Skeletal muscle accounts for ~80% of glucose uptake during an insulin clamp [18] and can increase from resting levels anywhere from 20-fold (~0.25 mM/min → ~4 mM/min single leg [19]) to 50-fold [20] during exercise [19,21,22]. This dynamic range for insulin-dependent and insulin-independent glucose uptake [21,23,24] has resulted in exercise therapies being touted as the most effective treatment strategies to combat the disease [14,25,26,27,28] and as a consequence skeletal muscle is a primary target in diabetic research.

Skeletal muscle mitochondria oxidize carbohydrates to support ATP production and fuel contracting muscle and as a consequence glucose uptake is largely driven by mitochondrial ATP production [1,3,29]. Exercise training improves skeletal muscle MOP capacity [30,31] with a concomitant increase in glucose tolerance [32]. Meex and colleagues reported a reduction in MOP capacity in diabetic subjects compared with healthy controls using phosphorus magnetic resonance spectroscopy (^31^PMRS). Following a 12-week exercise training regimen, MOP capacity was restored, and glucose tolerance improved. The close association of MOP capacity and glucose tolerance has resulted in a glut of hypotheses involving mitochondrial dysfunction in the development of type 2 diabetes.

Skeletal muscle mitochondrial dysfunction in type 2 diabetes was suggested by Kelley and colleagues in 2002 after reporting reduced electron transport chain activity measured from NADH:O2 oxidoreductase [33]. This study also reported a reduction in mitochondrial fractional volume measured from electron microscopy and citrate synthase activity. Several years later Shulman (2005) expounded upon the observations of Kelley and other investigators arguing that mitochondrial dysfunction was not only present in type 2 diabetes, but a major contributor to the development of insulin resistance [34]. Mitochondrial dysfunction in type 2 diabetes has gained widespread support since 2002 despite contradictory reports. This review summarizes the body of literature for both human and animal models and attempts a balanced summation of mitochondrial (dys) function in type 2 diabetes.

## 2. Mitochondrial Oxidative Phosphorylation (MOP)

The primary function of mitochondria is to provide the cell with ATP. While this organelle plays other roles that may contribute to cell survival (e.g., ROS signaling, calcium toxicity, apoptosis, etc., [35]), here only ATP production is used to define the function as a conceptual simplification. Mitochondria fuel oxidative phosphorylation by consuming oxygen at cytochrome c oxidase (complex IV). This is a complex process that begins with fuel conversion by the tricarboxylic acid (TCA) cycle to NADH and FADH_2_ to be used by the electron transport system (ETS) ultimately generating ATP (Figure 1) (for detailed description see Bioenergetics by Nicholls and Ferguson [36]). 

### 2.1. Tricarboxylic Acid Cycle

The TCA cycle resides within the mitochondrial matrix and produces reducing equivalents for the ETS. Although the TCA cycle is complex, for brevity it will be described only in the production of reducing equivalents to permit ATP production and evaluate mitochondrial function (Figure 1). Composed of nine enzymes, the cycle begins with citrate synthase catalyzing the condensation of acetyl-CoA with oxaloacetate and water to produce citrate. Acetyl-CoA is introduced from carbohydrate or fatty acid oxidation from pyruvate dehydrogenase (PDH) or 3-ketoacyl-CoA thiolase reactions respectively. Within the cycle these biochemical reactions produce reducing equivalents including NADH and FADH_2_ for the use at complex I (CI) or complex II (CII) to produce ubiquinol. During the cycle three NAD^+^ are reduced to NADH; one each by isocitrate dehydrogenase, alpha-ketoglutarate dehydrogenase, and malate dehydrogenase and one FAD^+^ is utilized by succinate dehydrogenase to produce FADH_2_ and subsequent QH_2_. Oxidation of carbohydrates produces an extra NADH from the PDH reaction and fatty acid oxidation produces an additional NADH and FADH_2_ from the 3-hydroxyacyl-CoA dehydrogenase reaction and acyl-CoA dehydrogenase respectively. This results in a total of 4:1 NADH:QH_2_ produced per acetyl-CoA from pyruvate oxidization and 4:2 NADH:QH_2_ per acetyl-CoA from fatty acids. 

Flux through the TCA cycle is regulated in part by the redox potential and phosphate potential to maintain the energetic status of the cell. This regulation is described using a series of reservoirs as metaphors for the potential energy differences by Meyer and Wiseman (2011) [3] and demonstrate that metabolic flux is driven by the cytosolic ATP demand (i.e., increased $\frac{ADP+Pi}{ATP}$ and $\frac{NAD^{+}}{NADH}$). It follows that TCA cycle flux also must increase with increasing ATP demand and this results in increased glucose oxidation to provide substrates to the TCA cycle. This fundamental principle of metabolism serving as a pull-through system was demonstrated by Jensen and colleagues showing a direct increase in glucose oxidation with increased rate of ATP consumption (ATPase rate). Escherichia coli were transfected with the F_1_ portion of the membrane bound F_1_F_O_ ATPase to increase ATP consumption without affecting other aspects of metabolism. Glucose consumption through glycolysis directly increased with increased ATP consumption demonstrating the pull-through effect of ATP demand [37]. The relationship between ATP demand and glucose oxidation perhaps explains at least a portion of the improved glucose handling in type 2 diabetes following exercise intervention.

### 2.2. Electron Transport System

Reducing equivalents feed into the ETS to support chemiosmotic ATP production. NADH and FADH_2_ produced from the TCA cycle are incorporated into the ETS by the donation of electrons at NADH dehydrogenase (CI) or succinate dehydrogenase (CII), respectively. Donated electrons pass down an electrochemical gradient through membrane soluble carriers to complex III (CIII) and complex IV (CIV). The energy provided from donated electrons fuels proton pumping across the mitochondrial inner membrane and develops a proton motive force (Δpmf, resting value 180 – 200 mV) that is a function of the membrane potential (~170 mV) and proton concentration gradient (10-30 mV) (Figure 1). This Δpmf permits ATP production by ATP synthase (F_1_F_O_ ATP synthase) as originally described by Mitchell in his “chemiosmotic theory” [38]. During increased ATP demand imposed by exercise, the proton gradient is dissipated through the ATP synthase to produce ATP in a direct relationship with demand [3]. ATP is transported from the matrix to the cytosol through adenine nucleotide translocase (ANT) and this transport is the accepted limitation to ADP sensitivity [39,40]. ANT activity directly influences the phosphate potential and reductions in activity are frequently suggested as the cause of impaired ATP production. However the precise mechanistic role of ANT in metabolic diseases including diabetes remains to be established empirically [41,42,43,44]. 

Oxidation of NADH and transfer of electrons to CI provides the energy to pump 4 protons across the inner membrane contributing to the proton motive force. In contrast, CII oxidation of FADH_2_ does not pump any protons but is used to produce QH_2_ [36]. The electrons produced at CI and CII transfer through the ETS via QH_2_ to CIII which pumps 2 protons. Finally, electrons are transferred via cytochrome c to CIV where two molecules of cytochrome c are oxidized to reduce $\frac{1}{2}$ O_2_ to water, and 4 protons are pumped across the inner membrane. Together, this results in a total of 10 protons pumped per NADH and 6 protons per FADH_2_. The F_1_F_O_ ATP synthase utilizes this proton gradient by coupling the backflow of protons into mitochondrial matrix with the synthesis of ATP at the cost of 8/3 protons per ATP synthesized. The non-integer proton value originates from the fact that 8 c-subunit rings are associated with each ATP synthase molecule in mammals, and a complete 360° rotation of the F_1_ catalytic head produces 3 ATP molecules [45]. Accounting for an additional proton used for ATP and Pi translocation across the inner membrane by ANT and the inorganic phosphate carrier (PiC), respectively, this results in the calculated ATP:O ratios of 10/(8/3+1), or 2.73 for NADH and 6/(8/3+1), or 1.64 for FADH_2_. Thus, the amount of ATP produced per O_2_ is dependent on which substrates are used to energize mitochondria. 

Production of ATP through the ETS, like TCA cycle regulation, is driven by ATP consumption via feedback regulation of ADP, Pi, and ATP. Other controllers and influences on respiration have been proposed and suggested to alter metabolic flux [46,47,48], however none occur without ADP and Pi being the prominent controllers of respiration. The result of this tight regulation permits maintenance of the intracellular free energy of ATP hydrolysis that ultimately determines the energy released from ATP hydrolysis. This intricately linked system comprises the ATP production capacity or MOP capacity of the mitochondria that is routinely quantified experimentally.

## 3. Measurement of Mitochondrial Function

MOP capacity is a combination of two factors: mitochondrial respiratory capacity, and the absolute fractional volume of mitochondria present within a given tissue. Therefore, before concluding mitochondria are either functional or dysfunctional both components must be known. V_MITO_ is routinely measured through biochemical assays of isolated tissue (see Larsen et al. [2]) and for this reason the present review focuses on measures of overall MOP capacity and inherent respiratory function. Mitochondrial oxidative phosphorylation capacity involves ATP production from TCA cycle and ETS flux and is routinely measured both in vitro and in vivo (Figure 2). 

### 3.1. Mitochondrial Function Quantified In Vitro

Mitochondria isolated from skeletal muscle via centrifugation [48] allows study of the properties of the organelle in a well-controlled external environment [49]. Controlling in vitro conditions permits quantification of mitochondrial biochemical properties such as proton leak and the absolute maximal ATP production rates using non-physiological conditions that are never attained in vivo because the muscle fatigues long before this can occur. Isolated mitochondria studies can be particularly valuable in metabolic diseases where it may not be known how excess glucose, lipids, or altered blood flow affect the mitochondrial respiration in vivo. Removing these factors by isolating the mitochondria permits direct quantification of mitochondrial respiratory capacity in vitro [50]. Instruments used to measure respiration in isolated mitochondria primarily include Oroboros O2k (Oroboros Instruments) and Seahorse XF Analyzers (Agilent). In the present review, the Oroboros O2k is primarily focused on because of its advantage in accuracy relative to Seahorse XF Analyzers which are designed for high-throughput qualitative measurements [51]. 

Mitochondrial function is quantitatively measured using high-resolution respirometry (Oroboros Instruments) to record changes in dissolved oxygen with a clark-type oxygen electrode. These changes in dissolved oxygen are direct indicators of mitochondrial oxygen consumption at cytochrome c oxidase. Mitochondrial function is determined by a series of metabolic challenges to quantify the leak state, maximal-ADP stimulated state, and respiratory control ratio (RCR, $\left[\frac{maximal\;ADP-stimulated\;state}{leak\;state}\right]$). The leak state is attributed to dissipation of the proton gradient across the inner membrane without ATP production. The leak state is determined in the chamber by presence of only mitochondria and substrate without ADP. Substrates for these studies generally include either pyruvate for carbohydrates or palmitoyl-CoA for fats. The maximal ADP-stimulated state is an absolute measure of maximal mitochondrial ATP production rates. To measure this a saturating bolus of ADP is added and all oxygen consumed is directly attributed to the ATP produced. Taken alone the maximal ADP-stimulated state is used to indicate function; however, it does not account for mitochondrial volume within a sample. For this reason, a known quantity of mitochondria is typically loaded into the chamber as determined from mitochondrial protein quantification. The RCR is the most widely accepted measure for mitochondrial function since this relative measure will not change with differences in mitochondrial sample loading [50]. There is not a set “standard” value for RCR, however when comparing between animals *within the same muscle group* it permits direct comparison of mitochondrial efficiency to produce ATP from ADP. RCR comparisons must be from the same type of tissue and using the same substrate since different substrates or tissues will change both measures of leak and the maximal ADP-stimulated state. For example, RCR values using pyruvate-based substrate are >20 in skeletal muscle, >17 in cardiac muscle, and >3–4 in liver while fatty-acid based substrates produce values of >8, >10, and >2–3 respectively [49,52,53]. In isolated mitochondria, measures of respiratory control are the most widely accepted indicators for mitochondrial function [50]. If RCR values fall below the expected range, the mitochondrial preparation should not be used as the isolation process caused a portion of the mitochondria to lose the tight coupling required for proper function. For this reason, RCR should always be conducted at the start of this type of experiment to verify the integrity of the inner mitochondrial membrane.

Measures of isolated mitochondria are performed in non-physiological conditions using saturating substrate and O_2_ but permit a direct comparison of mitochondrial respiratory capacity ex vivo. Maximal mitochondrial respiratory capacity measured ex vivo is well beyond the range of conditions that skeletal muscle mitochondria operate during normal duty cycles of contractile activity but is an important measure for objective quantification of mitochondrial function. Further measures to better understand in vitro respiration is not covered here but are well described by Fisher-Wellman et al. [54]. 

### 3.2. Mitochondrial Function Quantified In Vivo

Skeletal muscle permits the quantification of MOP capacity because ATP demand can be controlled and quantified in discreet steps through graded exercise intensities. Since ATP production is driven by ATP consumption, measuring these together is crucial to determine how well the mitochondria respond to step changes in demand (e.g., changes in cytosolic [ADP]). The study of skeletal muscle metabolism is powerful because ATP synthesis is driven by ATP demand and metabolic demand is difficult to quantify in other tissues in vivo. MOP capacity is routinely assayed by near-infrared spectroscopy (NIRS) by measuring hemoglobin and myoglobin desaturation [55,56,57,58] and by phosphorus magnetic resonance spectroscopy (^31^PMRS) measuring of phosphate metabolites [1,29,59,60,61,62,63]. In the present review, ^31^PMRS is the primary method discussed because of its quantitative advantage measuring metabolites inherent to mitochondrial respiration while NIRS quantification of oxygen dynamics is an indirect measure of mitochondrial function.

Using in vivo magnetic resonance MOP capacity is estimated by measuring the dynamics of the coupled metabolite phosphocreatine (PCr). Utilizing the creatine kinase equilibrium, changes in PCr directly reflect changes in ATP consumption/production [1,3,29,64,65]: $$\mathrm{ATPase}:\to \;ATP\;\to ADP+Pi+\;\alpha H^{+}\;$$
$$\mathrm{CK}\;\mathrm{equilibrium}:\to ADP+PCr+H^{+}↔ATP+Cr\;$$
$$\mathrm{Net}:\to PCR+\;\beta H^{+}\to Cr+Pi\;\;$$ where α (0.6) and β (0.4) represent partial protons produced as a function of environmental variables such as temperature, ionic strength, pH, and the differences in pK_a_ of substrate and product [66]. Tight coupling of the PCr-Cr pool to the adenylate pool through CK equilibration results in a stoichiometric relationship between PCr and Pi such that their sum always remains constant. Moreover, the total phosphate pool thus remains constant in skeletal muscle over the time course of a typical experiment and is an important cross-check for proper quantification at rest and during contractions. Failing to show conservation of the phosphate pool can lead to improbable physiological conclusions [67,68,69]. MOP capacity in vivo is most commonly determined from measuring PCr recovery time following exercise. The veracity of this measure was illustrated by Paganini et al. [70] where the PCr recovery time constant was directly related to mitochondrial volume and hence MOP capacity. Following alteration of muscle MOP capacity by chemical treatment (reduce) or exercise training (increase) the PCr recovery time grew faster with increased MOP capacity [70]. This technique has been broadly used in studies performed on human subjects and animal models and is a well-accepted measure of MOP capacity and by extension has been used to infer mitochondrial function [1,70,71]. However, this measurement relies on several assumptions which are often ignored or ill-considered (Table 1):

Ascribing PCr recovery to mitochondrial function is no longer valid if any of the assumptions in Table 1 do not hold true [72,73,74,75,76,77,78,79]. For example, in the case of peripheral arterial disease (PAD), blood flow to peripheral muscles is reduced depending on the disease progression and can limit oxygen delivery to working muscles [80]. Therefore, it is likely that oxygen is limiting mitochondrial respiration in individuals with PAD, violating assumption #2 and making interpretation of PCr recovery in relation to mitochondrial function equivocal. Likewise if oxygen becomes limiting, assumption #3 would likely be violated since the muscle would become acidic and directly reduce muscle mitochondrial ATP production [74,75,76,78,79]. Perhaps most often violated, assumption #3 requires interventions to be within the aerobic range where glycolytic ATP production is not significant. For example, during high intensity contractions above the lactate threshold [81] glycolytic ATP production is apparent and makes interpretation of the PCr recovery time constant no longer a valid measure of MOP capacity.

^31^PMRS can quantitatively determine mitochondrial function from several other perspectives (for review see Wiseman et al. 2008 [1]). Skeletal muscle experiments performed at multiple workloads where MOP ATP production can match ATP consumption permits quantification of apparent ADP sensitivity, free energy of ATP (ΔG_ATP_), ATPase rates from initial rate of PCr hydrolysis, glycolytic ATP production, and others (see [1]). For quantification of each of these components in skeletal muscle, experiments should include several work intensities within the sustainable range that an energetic steady state can be reached and abide by each of the assumptions outlined in Table 1. Results are no longer related to solely mitochondrial function if the assumptions are not valid and therefore these must be known especially in cases of metabolic disease where oxygen or substrate limitations have been suggested [4,13,59,82,83,84,85,86].

## 4. Obesity, Physical Inactivity, and Mop Capacity

MOP capacity can be altered by changes in f_MITO_, V_MITO_, or both. Mitochondrial fractional volume within skeletal muscle is highly plastic and sensitive to the effects of lifestyle including obesity and physical activity. Obesity and physical activity independently affect MOP capacity, through changes in mitochondrial volume (V_MITO_) but with some reports of reduced respiration per organelle (f_MITO_) [31,87,88,89]. This is particularly important in metabolic diseases, often termed “lifestyle diseases,” because over-nutrition and inactivity can play a role in their development. In fact, reduction in weight and increased physical activity vastly reduces the diabetes risk both independently and in conjunction with one another [90] and exercise continues to be the most effective treatment combatting type 2 diabetes [25]. 

### 4.1. Obesity Effect on MOP Capacity

Obesity affects MOP capacity as demonstrated in several models of murine species and in human subjects as well. In a detailed study by Turner et al. (2007) increased mitochondrial volume in mice and rats subjected to elevated fat intake were reported. C5BL/6J mice were acquired at 8 weeks of age and subjected to a high-fat diet for 20 weeks. Mice were sacrificed at 5 and 20 weeks of age and assayed for markers of mitochondrial volume including citrate synthase (mitochondrial matrix enzyme) and β-hydroxyacyl CoA dehydrogenase (β-HAD, enzyme of fatty acid oxidation). Feeding a high fat diet resulted in increased fat mass to obese levels and mitochondrial volume was elevated in quadriceps muscles by 20% in 5 weeks and 57% in 20 weeks. This result was also shown in the same study in Wistar rats subjected to 4 weeks of HFD, the chronically obese Zucker rat, and db/db mouse showing 14%, 36%, and 17% increased mitochondrial volume. Taken together, these observations show that increased fat composition leading to obesity can directly alter mitochondrial volume and by extension the MOP capacity across species [89]. 

The inherent effect of obesity on MOP likely depends on the stage of its progression since many other studies of obesity report contradictory effects on MOP capacity [88]. Mice subjected to a high fat and high sucrose diet for 16 weeks demonstrated reduced mitochondrial volume and maximal ADP-stimulated respiration [91]. Sprague Dawley rats subjected to a high fat diet for 3 weeks demonstrated increased ROS production and reduced maximal ADP-stimulated respiration [92]. The effect of obesity on MOP capacity also translates to humans where obese women demonstrated reduced mitochondrial function measured by reduced maximal ADP-stimulated respiration and reduced RCR in mitochondria isolated from vastus lateralis muscle [93]. Clearly, obesity can have a direct effect on MOP capacity, but the magnitude and direction of its effect are variable and thus requires careful matching for lean body mass in studies of type 2 diabetes. 

### 4.2. Physical Activity and MOP Capacity

Physical activity level directly influences skeletal muscle MOP capacity as shown by the late Dr. John Holloszy and colleagues in over 50 years of work studying exercise training [14,23,24,30,31,94,95,96,97,98]. In 1967, Holloszy reported a two-fold increase in mitochondrial fractional volume in rat muscles following a 12-week training regimen measured by the changes in cytochrome oxidase and succinate oxidase activities. Increased mitochondrial volume resulted in a two-fold increase in oxidative capacity measured from rat gastrocnemius and soleus muscles [31]. This was one of the first reports to demonstrate increased mitochondrial volume and muscle oxidative capacity together and was foundational to future exercise physiology studies. Among these adaptations, Holloszy hypothesized that increased MOP capacity from increased mitochondrial volume would increase the sensitivity of ADP feedback regulation [30]. In other words, less ADP would be required for ATP production as muscle mitochondrial volume increases. Thus, a given workload would disrupt the homeostasis less since ADP sensitivity is increased and increase muscle performance including less glycogen consumed and more work performed. This concept was subsequently supported in work by Chance and colleagues [99] and demonstrated by Terjung and Dudley [100]. Following alteration of muscle mitochondrial volume by exercise training (increase) or chemical treatment (decrease), Dudley et al. (1987) demonstrated that sensitivity to metabolic feedback is increased with increasing mitochondrial volume [100]. This higher ADP sensitivity enhances the maximal exercise performance (increased ATP production capacity) and endurance exercise performance (glycogen sparing [30]), a concept crucial to understanding muscle oxidative phosphorylation measured in vivo [1,70,101,102]. Since these early works, the relationship between training, mitochondrial volume, and MOP capacity has been well characterized in both the adaptable range of mitochondrial volume and its effect on MOP capacity [70,103]. 

More recently, studies have focused on the direct effect of sedentary behavior or physical *inactivity* on health and development of disease pathology. Inactivity is strongly related to development of obesity and insulin resistance [104]. In fact, sitting for more than ¾ of a day increases mortality risk by ~30% even in physically active individuals (greater than 7.5 METs per hour per week) [105]. The role of inactivity on disease progression in type 2 diabetes impacts several aspects of metabolism important in glucose handling [106,107,108]. After a brief period of inactivity (<5000 steps per day for 5 days) insulin sensitivity was reduced in healthy subjects measured from Matsuda index during an oral glucose tolerance test (OGTT), however there was with no change in blood flow response [107]. Using the same model of inactivity, during an OGTT arterial compliance was decreased and stiffness increased suggesting an effect of inactivity on vascular function contributing to insulin responsiveness also without reported changes in blood flow [106]. Finally, after comparable periods of inactivity reduced microvascular function was reported but without larger vessel dysfunction. This suggests that there is nearly immediate onset of microvascular dysfunction and insulin resistance with a sedentary lifestyle [108]. The impact of reduced microvascular function in progression of the disease particularly with respect to mitochondrial function and muscle performance have recently been suggested [4].

Sedentary lifestyle is directly related to a reduction in skeletal muscle MOP capacity. Booth and colleagues (1987) demonstrated 25% reduced MOP capacity after merely 7 days of sudden inactivity in rat quadricep muscle [109]. Both hindlimbs of female Sprague-Dawley rats (200–300 g) were immobilized by cast and muscle mitochondrial fractional volume was quantified from cytochrome c concentration after 7 days of immobilization. Subsequently, separate groups were immobilized for 7 days and casts removed to allow recovery from immobilization for 6 h, 2 and 4 days to directly relate results to sudden inactivity. Mitochondrial fractional volume measured by cytochrome c content gradually recovered reaching levels 94% of control in merely 4 days [109]. Taken together, these results (Figure 3), adapted from Morrison et al. (1987) demonstrate the direct effects of inactivity alone on mitochondrial fractional volume and MOP capacity.

The direct role of inactivity on MOP capacity also translates to human subjects [110]. Berg et al. (1993) showed 18% reduced mitochondrial volume in muscle biopsies from vastus lateralis muscle following 4 weeks of immobilization. A single hindlimb was immobilized by suspension with a harness and custom shoe for contralateral limb to ensure weight-bearing was null in harnessed limb. After 4 weeks of immobilization, mitochondrial fractional volume measured by citrate synthase activity was reduced (Pre: 40.2 ± 7.2 versus Post: 32.9 ± 4.9 μmol/g/min) with no change in the contralateral limb (Pre: 39.0 ± 4.7 versus Post: 37.1 ± 3.3 μmol/g/min) [110]. Berg et al. (1993) concluded that inactivity alone reduced mitochondrial volume and thus MOP capacity and similar results have been demonstrated following bedrest [111,112]. It is evident that inactivity by itself has a direct effect on MOP capacity and must be considered in metabolic disease especially those presenting comorbidities linked with inactivity [87,113,114,115]. In contrast, increasing activity vastly increases the quality of life in diseased individuals belabored by exercise intolerance [14,116] and suggests a direct role of activity on MOP capacity and disease progression. Future experiments must carefully match subjects for daily activity perhaps through questionnaires or wearing accelerometers although neither are perfect [117].

### 4.3. Summary

Obesity and inactivity each can independently influence skeletal muscle MOP capacity. Therefore, these variables must be controlled for especially in studies of “lifestyle diseases” including obesity, peripheral arterial disease, type 2 diabetes, or chronic obstructive pulmonary disorder. The role of further sequalae of obesity, inactivity, and metabolic syndrome that may alter apparent mitochondrial function including substrate and oxygen availability must also be weighed in the measure of MOP capacity and mitochondrial function (Table 1) but for brevity are not discussed here.

## 5. Human Type 2 Diabetes Mitochondrial Function

Mitochondrial function has been widely studied in human T2D, since the early works by Kelley and Shulman [33,118], who proposed disruption of aerobic metabolism as a mechanism in the development of this disease; however many studies contradict this view in the literature (Figure 4). Despite this, the dogmatic view that mitochondrial dysfunction plays a role in the development of type 2 diabetes is still widely accepted [119,120,121]. This section carefully reviews the main body of literature related to mitochondrial (dys)function in type 2 diabetes and concludes that many of these reported deficits are likely more related to obesity and sedentary lifestyles than the disease itself.

### 5.1. Human Type 2 Diabetes Mitochondrial Function In Vitro

Mitochondrial function in vitro is routinely measured in mitochondria isolated from skeletal muscle biopsies of individuals with type 2 diabetes. Several studies report reduced mitochondrial function [33,122,123,124] while others demonstrate no difference in function [125,126,127] when comparing obese controls with diabetic patients. Mogensen and colleagues (2007) reached the conclusion of mitochondrial dysfunction when mitochondria were isolated from vastus lateralis muscle of individuals with type 2 diabetes and their BMI-matched controls [122]. Oxygen consumption recorded polarographically (DW1 oxygraph) showed reduced maximal ADP-stimulated oxygen consumption rates when respiring in the presence of pyruvate and malate and no difference in presence of palmitoyl-carnitine and malate. However, neither daily activity nor muscle aerobic capacity were measured in these individuals [122], a limitation acknowledged by the authors in later publication [127]. Ritov et al. (2005) also reported mitochondrial dysfunction measuring 17% reduced ETS activity in diabetic patients compared to obese controls from vastus lateralis isolated mitochondria determined by succinate oxidase activity [124]. However, this was presented without accounting for the reported 27% reduction in mitochondrial fractional volume.

In contrast, normal mitochondrial function in muscle from individuals with diabetes compared to BMI-matched controls has been shown [125,127]. Hey-Mogensen and colleagues (2010) demonstrated nearly identical respiration in mitochondria isolated from vastus lateralis muscle of individuals with diabetes versus control subjects matched for age, physical activity, and BMI. Respiration was comparable in the presence of both pyruvate and malate as well as palmitoyl-carnitine and malate during the mitochondrial leak state and maximal ADP-stimulated state [127]. In a similar study, Abdul-Ghani and colleagues (2009) showed no difference between individuals with diabetes and BMI-matched controls. ATP production rates of isolated mitochondria were not different in the presence of either glutamate and malate or succinate and rotenone [125]. Taken together, evidence suggests that mitochondrial function is not reduced when isolated from diabetic patients and controlling for fractional volume, and further suggests no apparent link to the metabolic condition itself.

### 5.2. Human Type 2 Diabetes Mitochondrial Function In Vivo

Studies in type 2 diabetes have also reported mixed results when determining skeletal muscle mitochondrial function in vivo. Many investigators have suggested mitochondrial dysfunction [32,123,128,129] while others suggest none [115,130,131] when determined from PCr recovery time following exercise. Inspection of the results from these studies suggests differences are not inherent deficiencies of the mitochondrial organelle itself but rather deficiencies in study design.

Scheurmann-Freestone and colleagues (2003) demonstrated slowed PCr recovery in individuals with type 2 diabetes and concluded that MOP capacity was reduced because of mitochondrial dysfunction in type 2 diabetes [128]. In this study, individuals with type 2 diabetes were compared to healthy controls but factors including age, physical activity, and body mass index (BMI) were not weighed. Aging is frequently reported to independently contribute to reduced MOP capacity [114,132] and control subjects were ~10% younger than diabetic subjects. Further, type 2 diabetes frequently presents with obesity (87.5% of diabetic patients report overweight or obese [5]) and reduced physical activity (40.8% achieve less than 10 min of moderate physical activity per week [5]). The potential variability in mitochondrial function between subject populations is highlighted in work by Praet et al. (2006) [131]. In this work, PCr recovery times were measured in individuals with type 2 diabetes following exercise in the vastus lateralis muscle. Results demonstrated similar PCr recovery time constants to control subjects in the Scheurmann-Freestone (2003) study and the authors concluded no deficit in MOP capacity or mitochondrial function [131].

Subsequent studies also concluded mitochondrial dysfunction in type 2 diabetes from reduced PCr recovery time and controlled for obesity by BMI matching controls and diabetic subjects [32,129]. However, in each of these works physical activity levels were not quantified or compared between subjects and their respective mitochondrial fractional volumes were not measured. This leaves the possibility that the result could be explained by reduced MOP capacity because of sedentary lifestyle rather than the diabetes itself. Further, it is important to note that the BMI descriptor for obesity has recently been challenged [133,134,135]. When age, obesity, and activity levels were carefully controlled, no difference was measured in PCr recovery rates with the conclusion of normal mitochondrial function in type 2 diabetes [115,136]. De Feyter et al. [136] recruited patients based on age, similar BMI, and activity levels. Diabetics presented 200% higher fasting glucose and nearly 50% higher HbA_1C_. Despite pronounced type 2 diabetes, no differences were measured in PCr recovery after contracting at workloads where muscle acidity remained negligible. This work by De Feyter et al. suggests that in vivo dysfunction reported in type 2 diabetes is likely a result of age, obesity, and/or physical activity and not the disease itself [136].

### 5.3. What Is Missing?

Contradictory evidence appears to leave the question of mitochondrial dysfunction in type 2 diabetes unanswered. However, the literature suggests that conflicting reports of mitochondrial function in type 2 diabetes are the result of subject variability and not the disease itself. To correct this shortcoming, studies that report changes in mitochondrial function in vivo should also have mitochondrial function evaluated in vitro in the same subject. Further, studies must expand upon work by De Feyter et al. performing on well-controlled subjects matching for physical activity, obesity, and abiding by the assumptions in Table 1 by measuring pH, oxygen supply, and others across several contractile intensities. Variability in lifestyles within human populations proves difficult for such experimental design and thus studies rely on animal models of type 2 diabetes that provide cogent control of lifestyle variants before carefully translating to human populations.

## 6. Rat Models of Type 2 Diabetes

Muscle metabolism has been extensively studied in rat models [46,53,62,137,138,139,140] that allow for precise control of experiments and closely resemble human muscle metabolism and composition [141]. For example, studies in human subjects rely on maximum voluntary contraction of the muscle which can be difficult to interpret because of inconsistencies in muscle recruitment [142]. In contrast, the use of rodents provides the advantage of simultaneous recruitment of all hindlimb muscle fibers via electrical pacing of the sciatic nerve. Further, maximal recruitment in the hindlimb together with simultaneous force recording of muscle contraction permits measures to be performed at the optimal length of the muscle (L_O_, determined from the length tension relationship at which the maximal force is obtained). L_O_ is crucial in studies of muscle metabolism since any deviation can vastly influence the metabolic demand [143]. Quantification of muscle metabolism in rat models that present with the type 2 diabetic metabolic condition can be used to determine its role in mitochondrial function.

Many animal models of type 2 diabetes are used to study various organ systems and their link to the disease. For the purposes of brevity, this review focused on type 2 diabetes skeletal muscle mitochondrial function, and the studies discussed are confined to two of the most commonly used models including the Zucker Diabetic Fatty (ZDF) and Goto-Kakizaki (GK) rat models. For reference, Table 2 (adapted from work by Srinivasan et al. (2007)) [144] presents a summary of further animal models used in the study of type 2 diabetes and how they relate to the human condition in presentation of the disease. While rat models do not directly recapitulate the progression of type 2 diabetes in humans, studies within similar metabolic conditions (hyperglycemia, insulin resistance) can provide needed insight into the skeletal muscle mitochondrial function in type 2 diabetes.

### 6.1. Zucker Diabetic Fatty (ZDF) Rat

The Zucker diabetic fatty (ZDF) rat is perhaps the most widely used rat model for the study of the impact of type 2 diabetes. ZDF rats were developed through selective breeding of obese Zucker rats (OZR) showing the greatest increase in the severity of insulin resistance and poor glycemic control with time [145]. OZR manifest a dysfunctional (based in a coding error) leptin receptor and resulting in a severe leptin resistance, and the ensuing hyperphagia, obesity, and insulin resistance. However, because of insulin overproduction, most OZR do not present with severe fasting hyperglycemia until they become significantly older than matched ZDF [146]. It is important to note that OZR and ZDF rats are not equivalent animal strains and thus cannot be interchanged when comparing either existing studies or in experimental design and data interpretation for ongoing studies.

The OZR develop metabolic disease along a very predictable time course spanning many weeks. It begins with rapid increases in body mass until they are overweight/obese and gradually worsening glycemic control culminating in the development of a moderate level of hypertension [147] and the other systemic pathologies of “metabolic syndrome” [148]. Animals are exposed to these conditions over protracted periods of time prior to the overt development of end-organ damage and the consequences of long-term loss of glycemic control. In contrast, the ZDF rat model demonstrates a modest development of obesity, but a rapidly accelerated progression through insulin resistance to severe type 2 diabetes. Associated with this development, the ZDF exhibits a rapid progression of end-organ damage including early pancreatic and renal failure. Based on the dramatic differences in the physiology of these animal models caution should be exercised when interpreting the results of experiments using the different animal strains particularly regarding age, severity of metabolic dysfunction, and the potential influence of untoward pathology not part of the specific study. Mitochondrial function has been measured in the ZDF rat both in vitro and in vivo to determine whether (dys)function is present in T2D.

#### 6.1.1. Mitochondrial Function in Vitro

Skeletal muscle mitochondrial function quantified in vitro in the ZDF rat shows no difference in mitochondrial function. Mitochondria have been isolated from the skeletal muscles of ZDF rat tibialis anterior [140,149,150,151] and gastrocnemius muscles [152,153,154]. To date published results reveal no detectable deficiencies in mitochondrial respiratory capacity utilizing either carbohydrate (pyruvate and malate) [140,150,151,153] or fatty acids (palmitate or palmitoyl-carnitine and malate) [140,153,154] as substrate. However, Lenaers et al. reported reduced fat respiratory capacity at 6 weeks of age with no difference seen at 12 or 19 weeks [140]. Many other studies report an increase in fat respiratory capacity in ZDF compared with lean controls [149,150,151,152]. Wessels et al. (2014, 2015) reported no differences in isolated mitochondria but from animals that already had evidence of mitochondrial dysfunction measured in vivo by ^31^PMRS [150,151]. Taken together the results from the in vitro ZDF literature suggest that apparent mitochondrial dysfunction in vivo (if any) is likely a result of substrate or oxygen limitations or reduced mitochondrial volume fraction, since mitochondrial respiratory capacity is unaltered when studied in isolation.

#### 6.1.2. Mitochondrial Function in Vivo

ZDF rat mitochondrial function in vivo studies have reported inconsistent results. Wessels et al. (2014, 2015) described slowed skeletal muscle PCr recovery rates compared to age-matched lean controls and reported mitochondrial dysfunction in two separate studies [150,151]. Measurements were obtained from the tibialis anterior muscle following tetanic contractions (80Hz, 100ms, 1/s) that were designed to drive PCr concentration down to ~50% of resting levels (2-min stimulation). However, experiments performed with the same experimental protocol previously resulted in no difference in PCr recovery rates in ZDF rats and lean controls suggesting mitochondrial function is normal at ages 6, 12, and 18 weeks [136]. Differences in results likely arise from the protocol using 1 tetanic contraction per second which elicits a very intense metabolic demand nearly double the reported MOP capacity of the rat hindlimb of ~0.5 tetani/s twitch stimulation [155]. Interpretation of PCr kinetics beyond MOP capacity is arbitrary since oxygen delivery and pH changes are likely altered [73,155,156] and assumptions (2) and (3) from Table 1 are violated. Increased muscle acidity is directly proportionate to lactate production from glyco(geno)lysis [3,157,158,159] and PCr kinetics are no longer related to MOP capacity when the muscle becomes acidic [72,73,79]. Forbes et al. (2009) demonstrated a nearly two-fold slowing in PCr recovery time constant in rat gastrocnemius muscle measured at low intensity (0.75 Hz, 48 ± 3 s) and high intensity (5 Hz, 89 ± 8 s) twitch stimulation where acidosis was apparent and directly slowed PCr recovery in agreement with previous studies [74,75]. Taken together, the literature in ZDF rat skeletal muscle mitochondrial function in vivo is equivocal especially when performed at high intensities. Future experiments in these rats should include metabolic challenges across a broader dynamic range but within the MOP capacity (~<1 Hz twitch contractions).

### 6.2. Goto-Kakizaki (GK) Rat

The Goto-Kakizaki (GK) rat is the most widely used non-obese model of type 2 diabetes [160]. The spontaneously diabetic GK rat was developed through selective breeding of Wistar control (WC) rats. Oral glucose tolerance tests (OGTT) were performed on WC rats and the minority that demonstrated reduced glycemic control were inbred and this process was repeated over many generations resulting in the hyperglycemic and insulin resistant GK rat [161]. At a very young age, GK rats first present diabetic symptoms at around 3 weeks of age with hyperglycemia because of impaired beta cell insulin secretion [162]. Progression develops toward peripheral insulin resistance [163], hepatic glucose overproduction [164], and frank type 2 diabetes between 14–17 weeks of age [53,165]. GK rats demonstrate many of the comorbidities seen in human type 2 diabetes and thus are beneficial in application to human diabetes (for review, see [160,165]). Major advantages in studying the GK rat versus ZDF and others is this model presents without obesity [160] and is physically active [15,166]. This isolates the type 2 metabolic condition and permits a direct relationship between mitochondrial function and the type 2 diabetic metabolic condition itself.

#### 6.2.1. Mitochondrial Function in Vitro

Mitochondrial function in vitro in GK rat skeletal muscle shows findings are consistent with both human T2D and ZDF rat literature and shows no evidence of mitochondrial dysfunction. Skeletal muscle isolated mitochondria in the GK rat include data from both quadriceps muscles [52,167] and triceps surae (gastrocnemius, plantaris soleus) muscles [53]. These studies all report no deficit in skeletal muscle mitochondrial function measured by maximal ADP-stimulated respiration in the presence of varying substrates. Jørgensen et al. (2012) measured mitochondrial respiration in the presence of pyruvate and malate, palmitoyl carnitine and malate, and pyruvate, palmitoyl carnitine, and malate all together. Results showed no deficit in maximal ADP-stimulated respiration and an increase in capacity for 6 and 16 week old GK rats compared to 16 week old Wistar controls [52]. Similarly, Lai et al. (2017) detected no differences in GK rat skeletal muscle mitochondria when maximally ADP-stimulated at either 18 or 28 weeks in presence of glutamate [167]. Furthermore, Lewis et al. (2019) showed no detectable differences in isolated mitochondria in presence of pyruvate and malate and an increase in presence of palmitoyl carnitine and malate in the same group of animals that mitochondrial function was determined in vivo. Taken together, the body of literature for GK rat studies show no evidence of mitochondrial dysfunction ex vivo.

#### 6.2.2. Mitochondrial Function in Vivo

Skeletal muscle mitochondrial function in vivo has been measured by ^31^PMRS in the GK rat with the qualitatively consistent conclusion that mitochondrial function is not impaired [53,168,169]. Liu et al. (2016) report normal mitochondrial function in the GK rat from similar PCr recovery time compared to Wistar controls [168]. In this study the rat hindlimb was occluded by an inflatable cuff to cause ischemic depletion of PCr, and upon releasing the cuff PCr recovery time was measured. Although no differences were observed, PCr recovery following ischemia occurs in a very acidic environment violating a primary assumption regarding the utility of PCr kinetics and mitochondrial function (Table 1) [72,73,79]. This is demonstrated in Figure 3 of Liu et al. (2016) showing pH dropping to about 6.7 during ischemia and the authors acknowledging that this may influence the result [168].

In an elegant experiment, Macia and colleagues (2015) reported normal mitochondrial function in the GK rat with similar PCr contractile steady states and recovery time constants compared to Wistar controls [169]. Rats were stimulated at 3.3 Hz and ^31^PMRS was measured from the gastrocnemius muscle using a custom built NMR probe [139]. PCr kinetics were not different between groups during contraction despite reduced force production in the GK rat. The authors acknowledged this by stating that although force production was different, ATP cost per contraction was not different between groups and allowed direct comparison. These measures together are crucial for accurate quantification of mitochondrial function using ^31^PMRS and are often not an integral part of the experimental protocols. Although no differences were seen, 3.3 Hz is well beyond the MOP capacity of the hindlimb for twitch stimulation (reported as 1 Hz [155]) and makes interpretation of PCr recovery time constants for inferring mitochondrial function debatable [72,73]. To this point, PCr hydrolysis reached >80% of the resting concentration, pH dropped from ~7.1 to 6.2 - 6.4 and PCr recovery times were nearly 3X previously reported values even in controls [46,53,62,70] suggesting that this experimental system causes an occlusion of blood flow and therefore inadequate oxygen delivery.

In a separate study, PCr kinetics were measured in the GK rat and its Wistar control at several contractile intensities in the aerobic range [155] (where ATP consumption can be matched by mitochondrial ATP production) [53]. This range of challenges minimizes the risk of violating the primary assumptions in Table 1. Intensities of stimulation included 0.25, 0.5, 0.75, and 1 Hz and also above the aerobic capacity at 2 and 4 Hz to directly compare to previous work. Data showed a metabolic limitation above the aerobic capacity since more PCr was hydrolyzed during contraction in the diabetic animal. Paradoxically, there were no differences observed within the aerobic range which suggests that there is normal mitochondrial function. Together with in vitro data where mitochondria appeared fully functional in both diabetic and control animals, this suggested that limitations above the aerobic capacity were due to deficits unrelated to the mitochondria [53]. This result sheds light on rodents and human literature where metabolic limitations can be a result of factors that influence mitochondrial function that are apparent in type 2 diabetes including cardiovascular disease [15,170,171,172,173], vasculopathy [4,26,83,86,174,175], neuropathy [176,177], and more that may contribute to a reduced metabolic performance [4].

## 7. Summary and Future Directions

All available evidence in the literature from both human type 2 diabetes and rat models of type 2 diabetes suggests that skeletal muscle mitochondrial dysfunction is not inherently linked to type 2 diabetes. When isolated mitochondria are investigated using high-resolution respirometry the available literature suggests mitochondrial respiratory capacity is not reduced in type 2 diabetes and mitochondrial function is intact. Since there are no inherent deficits in the organelle itself, any compromises to aerobic respiration measured in vivo may likely be a result of either changes in mitochondrial fractional volume because of lifestyle differences or other factors in the cytosolic milieu such as oxygen limitations [4,70,178,179]. Vascular limitations have been well documented in type 2 diabetes [4,26,83,86,172,174,175]. To account for these variables the ideal experiment to determine the limitation of muscle metabolism and any potential mitochondrial role in a range of human pathologies requires quantification of these four contributors to aerobic respiration: 1) Mitochondrial fractional volume, 2) ATP demand, 3) ATP production, and 4) oxygen supply. The effects of oxygen limitations on mitochondrial ATP production were described by Connett et al. by compiling a large body of previous work and published in 1990 [178]. A modified figure from this work is presented here and describes the integration of each component in the ideal experiment (Figure 5).

Under normal circumstances ATP production is a function of ATP demand because the metabolic biproducts (ADP, Pi) of ATP hydrolysis drive the mitochondrial respiration. ATP demand is described along the x-axis in Figure 5 with increasing ATPase rates requiring increased supply of oxygen and is presented as a percent of the aerobic capacity dictated by muscle MOP capacity. The y-axis in Figure 5 is an indication of oxygen supply to respiring mitochondria; with increasing oxygen tension going up the y-axis. Within the green “Saturated” zone, mitochondrial respiration is not limited by oxygen supply and any reductions will not cause any changes in ATP production. This occurs because in saturating conditions, oxygen is not a controller of respiration, but only acts as a substrate in support of respiration [180]. For example, an individual in the early stages of developing peripheral arterial disease, oxygen supply to working muscle may be reduced because of reduced blood flow. However, no metabolic limitation (e.g., exercise intolerance) would occur as long as oxygen supply has not past beyond this theoretical threshold defined by the boundary between the green and grey zones. This boundary was empirically demonstrated by Rowell and colleagues in healthy individuals by altering arterial oxygen content [181]. Rowell et al. showed no deficit in muscle mechanical performance during arterial hypoxemia despite the likely reductions in cellular PO_2_ suggesting subjects did not fall out of the green “Saturated” zone. This concept may contribute to the difficulty in early diagnosis of PAD where patients’ symptoms only become evident after permanent damage has taken place [182].

If oxygen supply is limited enough for a given ATPase rate to fall within the grey “Metabolic Phenotype” zone, ATP production would not be limited but would result in a reduced phosphorylation potential. This means that more ADP would be present and more PCr would be hydrolyzed within this zone as a result of the reduced oxygen availability. Thus, although ATP production is not limited, this condition will present with a reduction in ADP sensitivity and mimic the result observed from changes in mitochondrial fractional volume [100]. Finally, if oxygen supply were further limited and fell within the red “O_2_ limited” zone ATP production would become oxygen dependent. This would present in vivo with increased PCr hydrolysis and slowed PCr recovery time constant but without any mitochondrial dysfunction.

Application of this conceptual framework demonstrates why therapeutic techniques that increase the mitochondrial fractional volume shift the sensitivity to ADP by changing the oxygen consumed per mitochondria [100]. In patients with COPD arterial oxygen supply limits muscle performance [183] and rarely can alveolar oxygen exchange be salvaged to improve this condition. However, exercise training has been shown to reduce lactate production and ventilation in COPD patients as a result of increased mitochondrial fractional volume [116] shifting these individuals closer to the green “Saturated” zone. Figure 6 illustrates some examples of different metabolic conditions and where their limitations to aerobic respiration may fall. Note that the boundary conditions between the grey and red zones define a critical point where mitochondrial function becomes oxygen dependent. This critical point will differ depending on the disease pathology and the inherent state of fitness of the subject.

The critical point where aerobic respiration becomes limited by oxygen supply was demonstrated many years ago by Stainsby et al. in 1964 using a canine gastrocnemius-plantaris muscle in situ [184]. In this study, muscle VO_2_ was chronically measured, while also systematically reducing either the blood flow or arterial oxygen content. Utilizing this technique, Stainsby et al. demonstrated that VO_2_ remains unaltered in contracting skeletal muscle until oxygen supply is limited by either reduction in blood flow or by reduction in oxygen content at the same blood flow rates. A similar experiment was designed utilizing the concept in Figure 5 by Lanza and colleagues in resting muscle [179]. In this work, ^31^PMRS was utilized to determine the critical point for ATP production prior to, during, and after inflatable cuff occlusion and measuring oxygen limitations by ^1^H measures of myoglobin desaturation [185,186]. Work in contracting muscle has been performed by McCully and colleagues [77] measuring the maximal respiration by ^31^PMRS and blood flow by doppler ultrasound. Blood flow was restricted in a stepwise fashion by cuff occlusion and the resulting changes in mitochondrial ATP production rates because of these reductions in blood flow were quantified. While these experiments were performed with another goal, these authors illustrated an experimental design that can be used to define the critical point in human subjects subjected to modest workloads.

Cree-Green and colleagues showed that oxygen could be limiting MOP capacity in type 2 diabetes patients utilizing ^31^PMRS [187]. PCr dynamics were measured following cessation of contraction with and without oxygen supplementation by changing oxygen concentrations of inspired air (FIO_2_). When breathing normal air (~21% O_2_), diabetics demonstrated reduced MOP capacity compared to controls and this reduction was alleviated when breathing 100% O_2_ [82]. This technique was previously demonstrated by Sunoo et al. in rats [188] and Richardson, Haseler, Hogan and colleagues in humans to change arterial oxygen content and alter phosphorus dynamics [84,189,190,191,192]. Depending on the imposed workload and state of physical fitness of the subject, changes in phosphorus dynamics were used to identify the limitation to aerobic respiration. In fact, this series of studies demonstrated limitations that fall within each zone of Figure 5 after reductions in oxygen supply showed: 1) No influence on phosphorus dynamics [84,190], 2) altered ADP sensitivity [191,192], and 3) slowed PCr recovery [84,189]. These conditions would fall within the “saturated,” “metabolic phenotype,” and “O_2_ limited” zones respectively.

Determining where the critical point falls for healthy individuals and type 2 diabetic patients requires a sophisticated experimental design to delineate the limitation of muscle metabolic performance. This concept has been described by Poole and colleagues as the theoretical “tipping point” where oxygen supply becomes limiting to mitochondrial respiration [193,194]. Although oxygen limitations in diabetes have been suggested, it has not been *quantitatively* determined if oxygen is in fact limiting ATP production. This must be performed by concurrent measurement of mitochondrial fractional volume, ATP production, ATP demand, and oxygen supply to respiring muscle. To illustrate this point we present an example of such an experiment in Figure 7 where a rat has been studied by magnetic resonance to measure ATP production and demand by ^31^PMRS together with measures of blood flow using ^1^H-based phase contrast angiography. In Panels D-F, as contractile intensity increases toward the aerobic capacity of rat hindlimb (~1 Hz twitch stimulation, [155]) ATP demand (initial rate of PCr hydrolysis described above) increases and a steady-state PCr is reached during contraction (ATP production=ATP demand). In panels A-C, blood flow (indicative of oxygen supply) increases in a reciprocal fashion as expected in a healthy rat. Utilizing this experimental design, within disease conditions such as type 2 diabetes, allows a quantitative understanding of any changes in oxygen delivery compared to healthy animals and where the “tipping point” may fall as contractile intensity is increased. As work intensity increases, the “tipping point” is approached and the reduced blood flow would cause increased PCr hydrolysis and recovery time constant. Once the “tipping point” is surpassed ATP production would no longer be able to match ATP consumption, a PCr steady state would not be reached, and PCr recovery time would increase. The limitation to this experimental design however is that it does not quantify O_2_ specifically at the level of the muscular bed. This has been vastly improved by getting closer to the blood:myocyte exchange of oxygen by Poole and colleagues utilizing phosphorescence quenching [195]. Other measures of oxygen supply (for review, see Sanjay et al. [196]) are improving non-invasive quantification but at present such measures have not developed for concomitant ^31^PMRS.

Review of available data suggests that in type 2 diabetes any deficit in MOP observed in vivo is likely vascular in nature provided there is proper matching of control subjects with their diabetic counterparts regarding lean body mass and daily activity levels. Future experiments must be designed to quantitatively determine such limitations to guide intervention strategies. Once these techniques are clinically approved, their application will vastly improve the understanding of diseases presenting with exercise intolerance and impaired activity of daily living. Cogent understanding will improve pharmacological and therapeutic interventions toward the goal of improving quality of life in these patients.

## Figures and Tables

**Figure 1 ijms-20-05271-f001:**
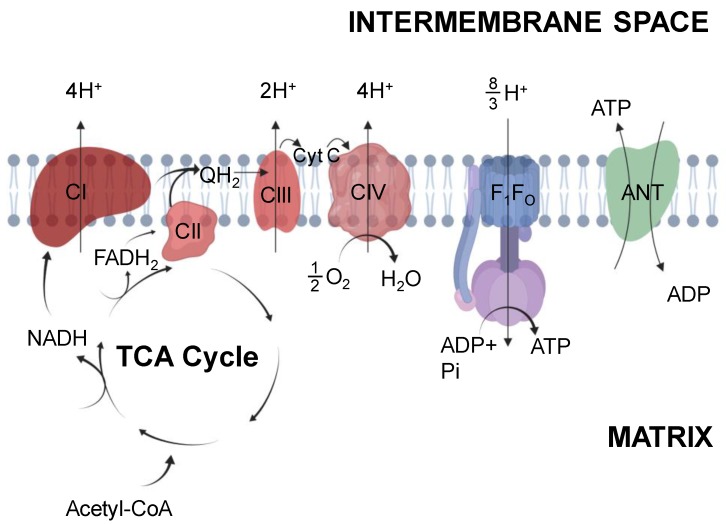
Mitochondrial oxidative phosphorylation. Mitochondria produce reducing equivalents through the tricarboxylic acid (TCA) cycle within the mitochondrial matrix. Reducing equivalents (NADH, FADH_2_) contribute to the electron transport system at complex I and II, respectively. Donation of electrons provides the energy to pump protons to generate the proton motive force (pmf). This pmf is utilized by the F_1_F_O_ ATP synthase in the production of ATP which is translocated to the intermembrane space through the adenine nucleotide translocase (ANT). The proton pumping stoichiometries are given as “effective” protons per pair of electrons. Created with Biorender.com.

**Figure 2 ijms-20-05271-f002:**
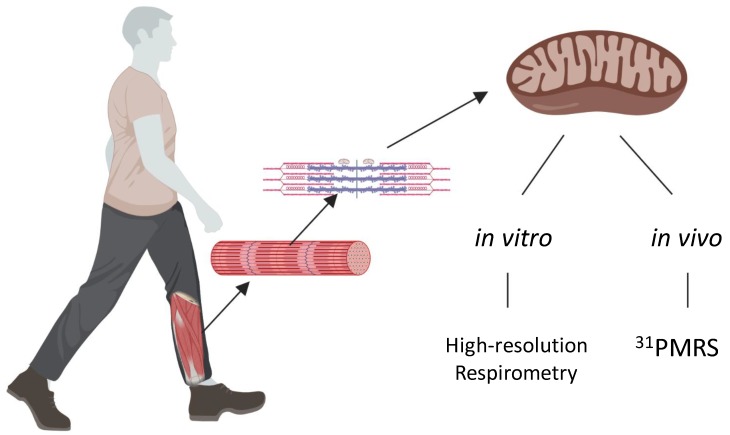
Mitochondrial oxidative phosphorylation capacity is comprised of both number and function that can be measured in vivo or in vitro. Mitochondria within skeletal muscle measured in vivo can be measured via phosphorus magnetic resonance spectroscopy (^31^PMRS). ^31^PMRS infers function from measures of mitochondrial oxidative phosphorylation (MOP) capacity, that when paired with V_MITO_ quantitatively describes function. Mitochondrial function in vitro can be measured via high-resolution respirometry following isolation of the organelle from the tissue. Created with Biorender.com.

**Figure 3 ijms-20-05271-f003:**
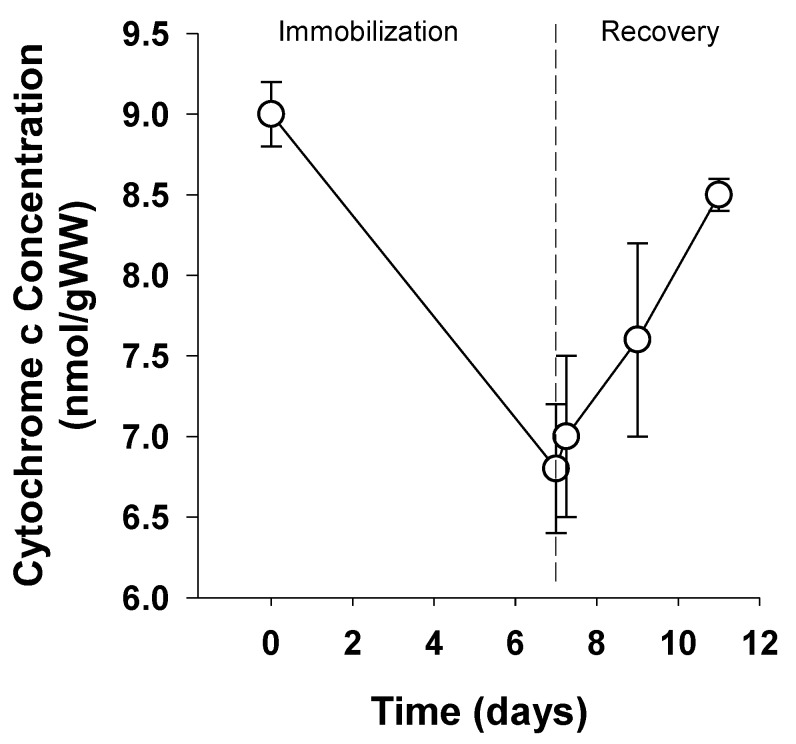
Role of inactivity on mitochondrial fractional volume adapted from Morrison et al. 1987. Rat quadricep muscles were immobilized by cast for 7 days and mitochondrial fractional volume determined from cytochrome c content. Measures were taken from muscle samples before immobilization and at time points 0-, 0.25-, 2-, and 4-days following cast-removal.

**Figure 4 ijms-20-05271-f004:**
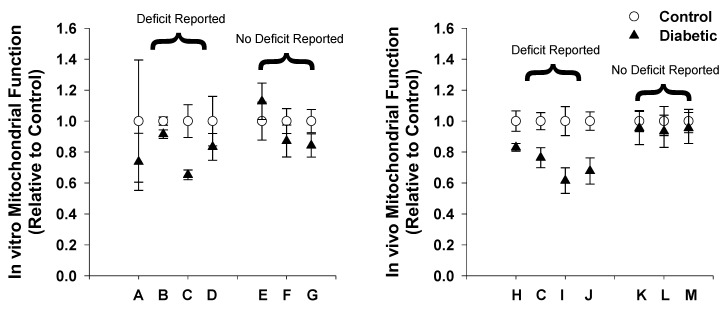
Summary of human mitochondrial (dys)function in diabetes. Data of reported mitochondrial (dys)function was compiled from the literature and presented relative to respective control subjects. Control (white circles) is always equal to 1 and diabetics (black triangles) presented as a fraction of that determined from the reported measures of function. This was determined for (A) in vitro and (B) in vivo studies where both deficits and no deficits in mitochondrial function have been reported. A) Kelley 2002; B) Mogensen 2007; C) Phielix 2008; D) Ritov 2005; E) Abdul-Ghani 2009; F) Boushel 2007; G) Hey-Mogensen 2010; H) Meex 2010; I) Scheuermann-Freestone 2003; J) Schrauwen-Hinderling 2007; K) De Feyter 2008; L) Praet 2006; M) van Tienen 2012.

**Figure 5 ijms-20-05271-f005:**
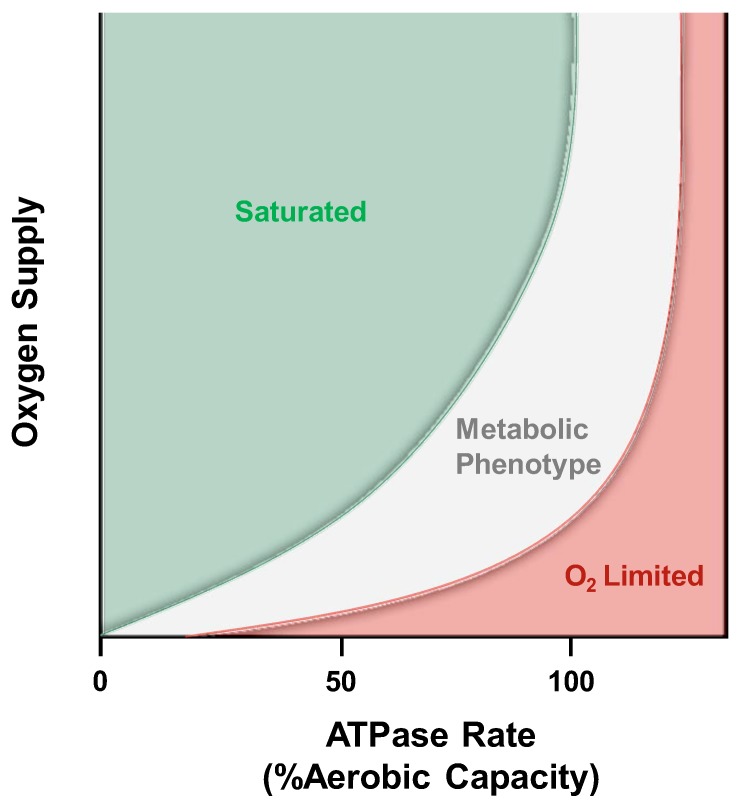
Theoretical description of the critical point for limitations to mitochondrial respiration. Modified from Connett et al. [178] this qualitatively shows the limitations to mitochondrial respiration. Axes are deliberately not defined, and the colored zones are conceptual rather than quantitatively meaningful. The green “saturated” zone represents where oxygen is in excess and reductions at low workloads will not affect mitochondrial function. If oxygen supply is decreased into the “metabolic phenotype” zone, mitochondrial respiration will not decline but at the expense of increased oxygen extraction and decreased phosphate potential. Finally, in the extreme or “O_2_ limited” zone, oxygen supply will limit mitochondrial respiration and result in a first-order dependence on oxygen that could be misinterpreted as mitochondrial dysfunction.

**Figure 6 ijms-20-05271-f006:**
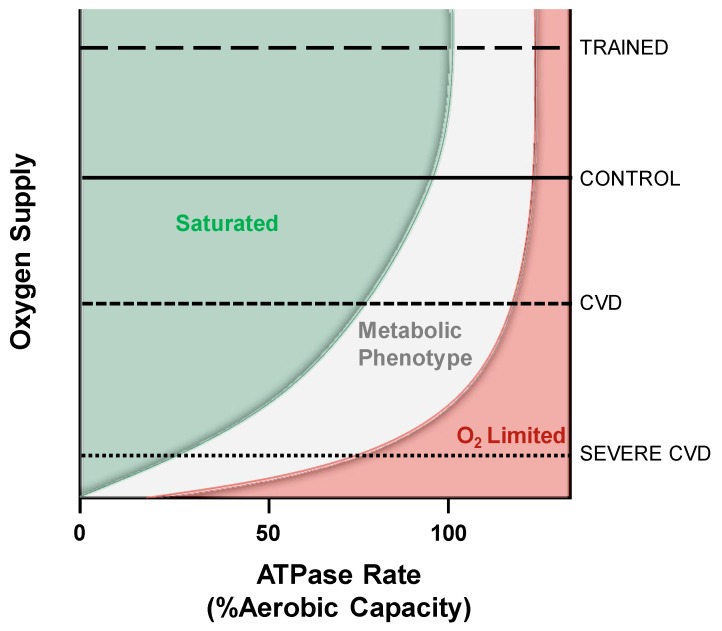
The critical point that limits muscle aerobic respiration varies between metabolic conditions. Within the control condition, oxygen is not limiting until the aerobic capacity is nearly reached and glycolysis affects the measures of MOP capacity inherent to mitochondrial function. As cardiovascular disease (CVD) develops, oxygen supply to muscle is reduced and results in a metabolic phenotype (reduced phosphate potential) at lower workloads that may develop into complete oxygen limitations in severe CVD. For simplicity here, severe CVD includes any disease presenting poor oxygen delivery including PAD, heart failure, COPD, and others. In contrast, exercise training increases mitochondrial fractional volume as well as oxygen delivering capacity and thus falls within a zone where oxygen saturation persists at higher workloads.

**Figure 7 ijms-20-05271-f007:**
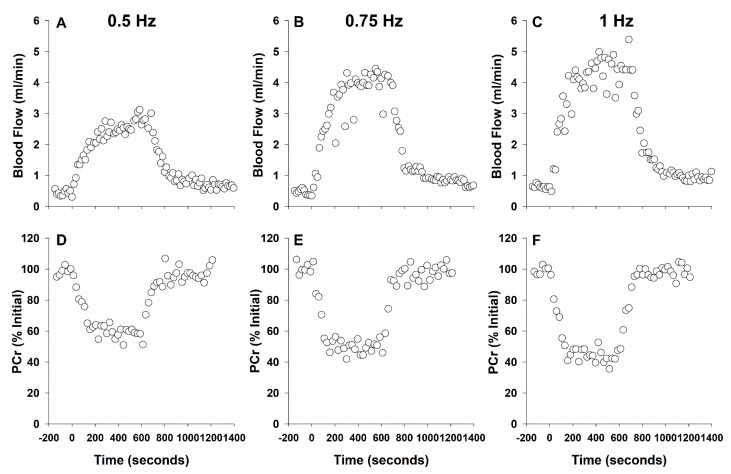
Experimental design measuring blood flow delivery and phosphorus energetics concurrently. As contractile intensity increases, blood flow to the muscle increases (**A**–**C**) and more PCr is hydrolyzed (**D**–**F**).

**Table 1 ijms-20-05271-t001:** Underlying assumptions for using the phosphocreatine (PCr) recovery time constant to infer mitochondrial function [70].

	Assumptions
1)	Equilibrium of the cytoplasmic creatine kinase reaction
2)	Oxygen and substrate supply are not limiting during recovery
3)	Glycolytic ATP production is negligible during recovery
4)	Mitochondrial ATP:O and basal muscle oxygen consumption (VO_2_) are each constant
5)	PCr resynthesis accounts for all but a negligible fraction of ATP consumed during recovery
6)	Similar mitochondrial fractional volumes within the measured tissues

**Table 2 ijms-20-05271-t002:** Summary of diabetic animal models re-summarized from Srinivasan et al. 2007.

	Model	Summary
**Genetically developed diabetic animals:**		
*Obese*:	ob/ob mouse	Pros: Develop resembling severe insulin resistance and type 2 diabetes in humans, homogeneous genetic background reduces variability. Cons: Highly genetic-determined diagnosis unlike mostly lifestyle disease in humans, limited and expensive, substantial maintenance often because of disease severity
	db/db mouse	
	Obese Zucker rat	
	ZDF rat	
	OLETF rat	
	Obese rhesus monkey	
*Nonobese*:	Cohen diabetic rat	
	GK rat	
	Akita mouse	
**Diet-induced diabetic animals:**		Pros: Diabetes progression similar to diabetes in obese populations overnutrition, toxicity from chemical induction avoided Cons: Generally, take a long time for development, frank hyperglycemia only apparent in certain animal models and therefore confounds development factors
*Obese:*	Sand rat	
	C57/BL 6J mouse	
	spiny mouse	
**Chemical, surgical-induced diabetic animals:**		Pros: Controlled onset of diabetic progression, residual insulin secretion better maintains health, inexpensive, recapitulates human with reduced beta cell mass. Cons: Hyperglycemia results mostly because of cytotoxic effect on beta cells versus initial insulin resistance, difficult for long term experiments because of beta cell regeneration, toxic actions on other organ systems
*Non-obese:*	HFD, low dose STZ rat or mouse	
	Neonatal STZ rat	
	Partial pancreatectomized animals	
**Myriad transgenic murine models targeting insulin receptors, glucose transporters, etc.**		Pros: Effect of single gene, better understanding of genetic contribution to type 2 diabetes. Cons: Highly expensive, translation to human population unclear

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
