# Peer review of "Quantification of Mitochondrial Oxidative Phosphorylation in Metabolic Disease: Application to Type 2 Diabetes"

_ijms, 2019, doi:10.3390/ijms20215271_

Round 1

Reviewer 1 Report

The paper presented by Lewis et al provides an extensive review on the contribution of the mitochondrial oxidative phosphorylation in type 2 diabetes and opens interesting perspectives on the evaluation and the impact of mitochondrial (dys)function on metabolic disorders.

The authors report in ordered detail the biological mechanisms underlying mitochondrial function and dysfunction, they dissect the experimental methods to asses mitochondrial function. The authors report on the animal models in an analytic approach. Finally, Authors provide a comprehensive discussion on the future perspectives.

Major comment: in Lines 309-330 the Authors reports the effects of obesity on MOP, the results are contradictory, Authors should comment.

Some minor text editing is needed:

Line 45: mismatching in the citation (Larson/Larsen)

Lines 77-80: the sentence should be checked

Line 107: error to be checked

Line 267: Paganini et al has reference number 68 not 124

Table 1: check format, not all the numbers are near the corresponding sentences

Line 361: OGTT, the abbreviation is cited before the full definition (line 584)

Line 377: error to be checked

Line 410: error to be checked

Line 512: error to be checked, Table 2 needs to be cited

Line 666: error to be checked

Line 710: error to be checked

Line 731: error to be checked, reference to figure 7?

Line 762: error to be checked, reference to figure 7?

Author Response

We thank the reviewers for their careful review of our work. We appreciate your time, effort and expertise and your suggestions have significantly improved the manuscript in substance and clarity. We have copied and pasted your comments and our responses follow in red font.

REVIEWER #1

The paper presented by Lewis et al provides an extensive review on the contribution of the mitochondrial oxidative phosphorylation in type 2 diabetes and opens interesting perspectives on the evaluation and the impact of mitochondrial (dys)function on metabolic disorders.

The authors report in ordered detail the biological mechanisms underlying mitochondrial function and dysfunction, they dissect the experimental methods to asses mitochondrial function. The authors report on the animal models in an analytic approach. Finally, Authors provide a comprehensive discussion on the future perspectives.

Major comment: in Lines 309-330 the Authors reports the effects of obesity on MOP, the results are contradictory, Authors should comment.

This is true and highlighting this was overlooked. It has been accounted for as below:

The inherent effect of obesity on MOP likely depends on the stage of its progression since many other studies of obesity report contradictory effects on MOP capacity [85]. Mice subjected to a high fat and high sucrose diet for 6 weeks demonstrated reduced mitochondrial volume and maximal ADP-stimulated respiration [88]. Sprague Dawley rats subjected to a high fat diet for 3 weeks demonstrated increased ROS production and reduced maximal ADP-stimulated respiration [89]. The effect of obesity on MOP capacity also translates to humans where obese women demonstrated reduced mitochondrial function measured by reduced maximal ADP-stimulated respiration and reduced RCR in mitochondria isolated from vastus lateralis muscle [90]. Clearly, obesity can have a direct effect on MOP capacity but the magnitude and direction of its effect are variable and thus requires careful matching for lean body mass in studies of type 2 diabetes.”

Some minor text editing is needed:

Line 45: mismatching in the citation (Larson/Larsen)

This has been fixed.

Lines 77-80: the sentence should be checked

This sentence had lost its meaning due to omitting the word “as”. This has been included and thank you for bringing this to our attention.

Line 107: error to be checked

Lost from wordàpdf. Have changed all figure references to ensure this cannot happen again.

Line 267: Paganini et al has reference number 68 not 124

Thank you for pointing this out and it has been fixed.

Table 1: check format, not all the numbers are near the corresponding sentences

We will bring this to the attention of the editor.

Line 361: OGTT, the abbreviation is cited before the full definition (line 584)

The first OGTT now follows its definition: ”oral glucose tolerance test”

Line 377: error to be checked

Line 410: error to be checked

Line 512: error to be checked, Table 2 needs to be cited

Line 666: error to be checked

Line 710: error to be checked

Line 731: error to be checked, reference to figure 7?

Line 762: error to be checked, reference to figure 7?

Each of these errors were a result of losing their links when transferred into the journal formatting. They have been replaced with the proper figure/table numbers.

Reviewer 2 Report

This a ponderous review about the involvement of mitochondrial function in the type-2 diabetes. This pathology is a metabolic illness and mitochondria are central in the energetic metabolism of the cell.

A large corpus of information is presented and extensive citations are also referred herein. The mitochondrial function is well outlined in normal and pathological conditions.

I would recommend to add in the introductory sections at least a short description of the hormones and effectors in the network controlling the mitochondria in cell subject to the deregulation of T2D.  

There are several warning in the pdf file about missing references and other.

Author Response

We thank the reviewers for their careful review of our work. We appreciate your time, effort and expertise and your suggestions have significantly improved the manuscript in substance and clarity. We have copied and pasted your comments and our responses follow in red font.

REVIEWER #2

This a ponderous review about the involvement of mitochondrial function in the type-2 diabetes. This pathology is a metabolic illness and mitochondria are central in the energetic metabolism of the cell.

There is no question that mitochondria are central to the energetic metabolism of the cell as we think this review describes.  The review attempts to cover a rather broad area of observations that have conflicting conclusions and we believe we need to draw the reader into the document and our logic to fully appreciate this conflict.  This does not mean that mitochondrial dysfunction is not causally related to diabetic symptoms within other tissues.  However, for  skeletal muscle we believe we firmly establish that it is more complicated than just the organelle itself.   There is no question that mitochondrial dysfunction could result in type 2 diabetes as has been shown in instances of mitopathies, however, this does not mean that type 2 diabetes requires mitochondrial dysfunction and it is the author’s opinion that this lack of distinction has largely cluttered the field on this subject.

A large corpus of information is presented and extensive citations are also referred herein. The mitochondrial function is well outlined in normal and pathological conditions.

Thank you.

I would recommend to add in the introductory sections at least a short description of the hormones and effectors in the network controlling the mitochondria in cell subject to the deregulation of T2D. 

We appreciate the breadth of your statement and indeed there are a lot of effectors such as PGC-1 that influence mitochondrial biogenesis but PGC-1 is also involved in VEGF expression and vascularization.  So, we believe that these are difficult to talk of separately and are the essential components of the tissue bioenergetic unit (capillaries and mitochondria) which has been extensively reviewed by Weibel and his co-workers over the past 3 decades.   Rather than delve into the myriad of potential effectors that impact this functional unit we chose to focus directly on the input (oxygen) and output (ATP).  However we thank you for this comment and the addition of these may provide a better context in their relation to T2D however there are many other considerations including inflammation-related mechanisms and hormonal effects as you suggest that we chose to refrain from discussing to keep the scope and depth reasonable.

There are several warning in the pdf file about missing references and other.

Each of these errors were a result of losing their links when transferred into the journal formatting. They have been replaced with the proper figure/table numbers.

These have been addressed.